

# A 3D particle Monte Carlo approach to studying nucleation

Christoph Köhn[1], Martin Bødker Enghoff[1], and Henrik Svensmark[1]

[1]Technical University of Denmark, National Space Institute (DTU Space), Elektrovej 328, 2800 Kgs Lyngby, Denmark

*Correspondence to:* C.K. (koehn@space.dtu.dk)

**Abstract.** The nucleation of sulphuric acid molecules plays a key role in the formation of aerosols. We here present a three dimensional particle Monte Carlo model to study the growth of sulphuric acid clusters as well as its dependence on the ambient temperature and the initial particle density. We initiate a swarm of sulphuric acid molecules with a size of 0.15 nm with densities between $10^7$ and $10^8$ cm$^{-3}$ at temperatures of 200 and 300 K. After every time step, we update the position and velocity of particles as a function of size-dependent diffusion coefficients. If two particles encounter, we merge them and add their volumes and masses. Inversely, we check after every time step whether a polymer evaporates liberating a molecule. We present the spatial distribution as well as the size distribution calculated from individual clusters. We also calculate the nucleation rate of clusters with a radius of 0.85 nm as a function of time, initial particle density and temperature. For 200 K, the nucleation rate increases as a function of time; for 300 K we observe an interplay between clustering and evaporation and thus the oscillation of the nucleation rate around the mean nucleation rate. The nucleation rates obtained from the presented model agree well with experimentally obtained values which serves as a benchmark of our code. In contrast to previous nucleation models, we here present for the first time a code capable of tracing individual particles and thus of capturing the physics related to the discrete nature of particles.

## 1 Introduction

Nucleation of aerosols is the fundamental process by which gas condenses to form stable clusters. These clusters can potentially grow all the way to sizes where they can serve as cloud condensation nuclei (CCN), typically 50-100 nm. The nucleation phenomenon has been observed all around the globe (Kulmala et al., 2004) and is also considered to contribute to the formation of clouds on brown dwarfs and exoplantes (Helling and Fomins , 2013; Lee et al. , 2015). About half of all CCN are estimated to originate from nucleated aerosols (Merikanto et al., 2009), making nucleation a relevant topic not only for aerosol research but also for its implications for cloud formation. Additionally both aerosols and clouds are relevant for e.g. climate change due to their large forcing effects (Boucher et al., 2013). The key molecule for aerosol nucleation has long been thought to be sulphuric acid (with water or other stabilizing molecules) due to its ability to form strong bonds (Curtius, 2006). Recently is has been shown that highly oxygenated organic molecules are also able to nucleate at high altitudes (Bianchi et al., 2016).

Traditionally nucleation has been described by classical thermodynamic nucleation theory (Hamill et al., 1982), kinetic numerical models (Pirjola and Kulmala , 1998; Lovejoy et al., 2004; Yu , 2006), or parameterisations based on either nucleation theory (e.g., Vehkamaki et al., 2002) or experimental data (e.g., Dunne et al. , 2016). The parameterisations and numerical





models have the advantage that they can be adapted for use in global modelling (Spracklen et al. , 2006; Yu et al. , 2008; Pierce and Adams, 2009) due to being computationally quick. Kazil and Lovejoy (2007) used a semi-analytical approach to add aerosols to a global model. More recently an Atmospheric Cluster Dynamics Code (ACDC) has been developed solving the so-called birth-death equations, ordinary differential equations (ODEs) describing the temporal evolution of cluster densities of

a given size (McGrath et al. , 2012). The novelty of such a model is the automatic generation of ODEs for a given cluster size and its implementation into the solver whenever needed. This approach aims to reduce typographical errors when implementing ODEs manually. Such traditional numerical models can provide information with regards to particle size distributions and may reflect the physics satisfactory using actual data for condensation and evaporation. The information provided by these models are, however, focused on developments in time and not in space. If each molecule could be tracked in a three dimensional space

it might be possible to achieve new insights into the process of nucleation.

Tracing individual particles in space and time is the main advantage of particle Monte Carlo codes. They are for example widely used to simulate the properties of lightning discharges (Chanrion and Neubert , 2008; Köhn et al. , 2017a, b) by tracing individual electrons and photons or to study the nanostructure growth of atoms on surfaces in electrochemical models (Jensen et al. , 1994a, b; Fransaer and Penner , 1999).

Monte Carlo models give the opportunity to include all relevant microphysical processes as well as the interaction amongst all involved particles. In contrast to kinetic models or pure parametrisations which are based on averaged quantities, such as the density or concentration of particles or the mean energy, these models are able to capture rare events initiated by single particles (Rubino and Tuffin , 2009; Hsieh , 2002) as for example the production of gamma rays or positrons in the vicinity of lightning discharges (Köhn and Ebert , 2015).

The disadvantage of Monte Carlo codes is their runtime. Depending on the size of the problem Monte Carlo simulations can take up to several weeks whereas models based on averaged quantities take several hours to days Li et al. (2012). For small systems, however, the time difference is not significant and Monte Carlo models offer a much better approach to the discrete nature of particles.

We here present a particle Monte Carlo code to study the nucleation of sulphuric acid clusters in 3D, which to our knowledge

has not been done before. We here emphasize, however, that we not only present a new model to simulate the nucleation of sulphuric acid clusters, but also that the implemented physics is sufficient enough to study the nucleation of neutral sulphuric acid clusters in order to benchmark our model. Previously Monte Carlo studies have been used on the sulphuric-acid water system to study cluster parameters such as cluster shape, conformation and dissociation (Kusaka et al. , 1998) as well as cluster free energies (Kathmann and Hale , 2001). The paper is organized as follows. In section 2 we introduce the Monte

Carlo particle model for the study of nucleation processes and discuss all the ingredients of this model: the implementation of single particles, diffusion coefficients, the collision of particles, the evaporation coefficients as well as the choice of the time step. In section 3 we discuss the spatial and size distribution of single sulphuric acid clusters and compare nucleation rates calculated with the present model with values from literature. We finally conclude in section 4.





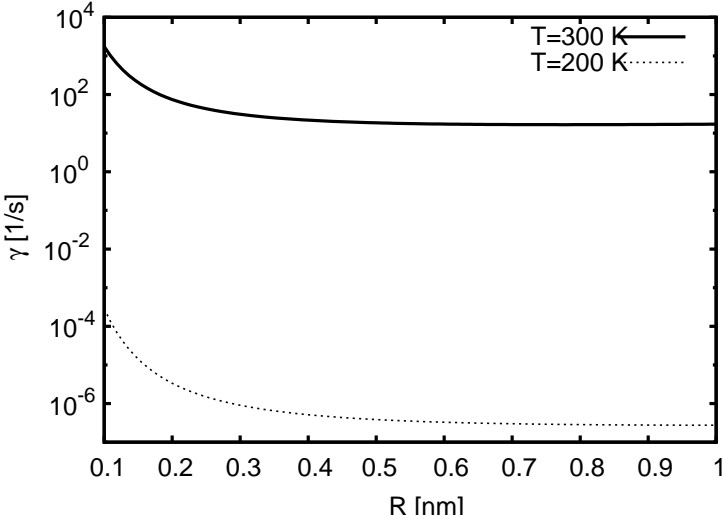

**Figure 1.** The evaporation coefficient $\gamma$ (7) for temperatures of 200 K and 300 K as a function of size $R$.

## 2   Modelling

For the study of the growth of $H_2SO_4$ clusters, we introduce a particle Monte Carlo code tracing individual $H_2SO_4$ molecules and clusters. In order to benchmark this model, we perform simulations with initial sulphuric acid molecule densities of $n = 10^7$ $cm^{-3}$ and of $10^8$ $cm^{-3}$ at 200 K and 300 K.

5    In the simulations we do not distinguish between molecules and clusters, we simply refer to them as particles. Each particle is described as a sphere characterized by its position $\mathbf{r} = (x, y, z)$ in Cartesian coordinates and its radius $R$.

After every time step $\Delta t$, the position is updated through

$$\mathbf{r}(t + \Delta t) = \mathbf{r}(t) + \sqrt{2D(R)\Delta t}\,\mathbf{G} \tag{1}$$

where $\mathbf{G} = (\varrho \cos\phi \sin\theta, \varrho \sin\phi \sin\theta, \varrho \cos\theta)$ is a Gaussian random number (Fransaer and Penner , 1999) with $\varrho = \sqrt{-2\log(r_1)}$, 10    $\phi = 2\pi r_2, \theta = \pi r_3, r_i \in [0, 1)$. $D(R) = D_0 \cdot (R_1/R)^2$ depends on the particle size $R$ and on the diffusion coefficient (Durst , 2006)

$$D_0 = \frac{2}{3}\sqrt{\frac{k_B^3 T^3}{\pi^3 m_1}}\frac{1}{4PR_1^2} \tag{2}$$

for molecules where $R_1 = 0.15$ nm Kuczkowski et al. (1981) and $m_1 = 1.6366 \cdot 10^{-25}$ kg are the initial size and the mass of single sulphuric acid molecules without any attached water molecules. $k_B \approx 1.38 \cdot 10^{-23}$ J/K is the Boltzmann constant 15    and $P = 1$ bar the ambient pressure. For $T = 300$ K and $T = 200$ K the diffusion coefficients are $D_0 \approx 10^{-5}$ m²/s and $D_0 \approx 5.4 \cdot 10^{-6}$ m²/s.



The time step used is related to the time it takes a particle to diffuse the average separation between particles. The average separation between particles is $\sqrt[3]{1/n}$ and the related diffusion length is $\sqrt{2D_0\Delta t}$. Equating the two gives

$$\Delta t \sim \frac{1}{2D_0\sqrt[3]{n^2}}. \tag{3}$$

Hence it is $\Delta t \sim 108$ $\mu$s for $n = 10^7$ cm$^{-3}$ and 300 K, $\Delta t \sim 199$ $\mu$s for $n = 10^7$ cm$^{-3}$ and 200 K as well as $\Delta t \sim 23$ $\mu$s

for $n = 10^8$ cm$^{-3}$ and 300 K. To ensure the comparability of all simulations and as a compromise of these three values, we therefore choose $\Delta t = 100$ $\mu$s.

After every time step we check whether two particles $i$ and $j$ with sizes $R_i$ and $R_j$ overlap by evaluating the condition

$$|\mathbf{r}_i - \mathbf{r}_j| \le R_i + R_j \tag{4}$$

where $\mathbf{r}_i$ are $\mathbf{r}_j$ are the particles' positions. If this condition is fulfilled, the two particles are merged by adding the mass and

volume of the two particles, hence $m_{i+j} = m_i + m_j$ and $R_{i+j} = \sqrt[3]{R_i^3 + R_j^3}$; the new position and velocity are determined through

$$\mathbf{r}_{i+j} = \frac{m_i\mathbf{r}_i + m_j\mathbf{r}_j}{m_i + m_j}, \tag{5}$$

$$\mathbf{v}_{i+j} = \frac{m_i\mathbf{v}_i + m_j\mathbf{v}_j}{m_i + m_j}. \tag{6}$$

Note that we add a new H$_2$SO$_4$ molecule to the simulation domain at a random position after merging two particles if the

particle number becomes smaller than the initial particle number; thus, the density of particles does not drop below the initial particle density.

Vice versa we also check after every time step whether a cluster with radius $R$ and mass $m$ evaporates by emitting one sulphuric acid molecule added to the simulation domain leaving a cluster with reduced mass and volume behind. The evaporation probability is $P_{eva} = \gamma\Delta t$ with evaporation frequency (Yu , 2005)

$$\gamma = \sqrt{\frac{8\pi k_B T(m_1 + m)}{m_1 m}}(R_1 + R)^2 n_{a,sol}^{\infty} \exp\left(\frac{8M_1 R_1^3 \sigma}{3m_1 \mathcal{R} T R}\right) \cdot \frac{R_1}{R} \tag{7}$$

where we have added $R_1/R$ to account for the size dependence of the concentration $n_{a,sol}^{\infty}$ of H$_2$SO$_4$ vapor molecules in the equilibrium vapor above a flat source; $n_{a,sol}^{\infty} \approx 6.4366 \cdot 10^{16}$ m$^{-3}$ and $n_{a,sol}^{\infty} \approx 9.7836 \cdot 10^8$ m$^{-3}$ are the concentrations for 300 K and 200 K, respectively (Seinfeld and Pandis , 2006, pp. 467–468). $M_1 = 98$ g mol$^{-1}$ is the molar moss of H$_2$SO$_4$, $\sigma = 76 \cdot 10^{-3}$ Nm$^{-1}$ the surface tension (Lange and Dean , 1967, pp. 1661–1665) and $\mathcal{R} \approx 8.31$ J (mol K)$^{-1}$ the universal gas

constant. Figure 1 shows the evaporation coefficient $\gamma$ as a function of size $R$ for 200 K and 300 K. For 300 K evaporation is most efficient for $R \lesssim 0.3$ nm and becomes less probable with increasing size. For 200 K $\gamma$ is several orders of magnitude smaller than for 300 K, thus compared to 300 K evaporation is negligible. Since the emitted molecule is added to the simulation domain, the particle density slightly increases; however, we have observed that it does not exceed $1.044 \cdot 10^7$ cm$^{-3}$ for an initial density of $10^7$ cm$^{-3}$ and $1.066 \cdot 10^8$ cm$^{-3}$ for an initial density of $10^8$ cm$^{-3}$.

In order to optimize the runtime of our simulations, we have chosen the volume of the simulation domain to be $10^{-4}$ cm$^3$ for a density of $10^7$ cm$^{-3}$ and to be $10^{-5}$ cm$^3$ for a density of $10^8$ cm$^{-3}$. Consequently we initiate all simulations by placing 1000 individual H$_2$SO$_4$ molecules at random positions into the simulation domain which scales with the concentration.





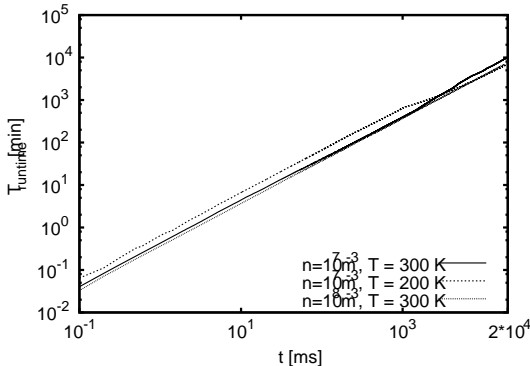

**Figure 2.** The runtime of simulations as a function of the simulated time $t$ for all considered cases.

## 3 Results

In the following we discuss the spatial distribution and the size distribution of all particles as well as the nucleation rates of clusters with a radius of 0.85 nm. As supplementary material we have added movies showing the temporal evolution of the particle position and size for $n = 10^7$ cm$^{-3}$ and 200 K as well as for $n = 10^8$ cm$^{-3}$ and 300 K.

Figure 2 shows the runtime of all simulations as a function of the simulated time. Except for some fluctuations, the runtime is rather independent of the initial particle density or temperature since we have chosen the same initial particle number in all simulations. The runtime increases linearly because of the approximate constancy of the number of simulated particles. In all considered cases it takes approximately $5 \cdot 10^3$ minutes or equivalently 3.5 days per 10 s of simulated time.

### 3.1 Spatial distribution

Fig. 3 shows the spatial distribution of all particles projected onto the $xy$ plane after 1 s and after 10 s. For better visibility we have multiplied the real size of each particle by a factor of $2 \cdot 10^6$. Note that the size of the simulation domain for $n = 10^8$ cm$^{-3}$ is smaller because of keeping the same initial molecule number as in the other two cases, but with a density ten times as large. Figure 4 illustrates the capability of the present model for full three dimensional simulations. It shows the spatial distribution after 10 s for the same densities and temperatures as in Fig. 3 d) and f). Fig. 3 and 4 demonstrate that in all cases the particles

are distributed randomly within the simulation domain. Panels a) and b) of Fig. 3 show that for $n = 10^7$ cm$^{-3}$ and $T = 300$ K, there is no significant growth between 1 s and 10 s; the average radius of all particles after 1 s and 10 s is approximately 0.151 nm. Similarly there is only a small difference between 1 s and 10 s for $n = 10^7$ cm$^{-3}$ and 200 K (c,d). However, in contrast to a temperature of 300 K where evaporation is not negligible, some clusters have grown further after 10 s. The average size is 0.157 nm after 1 s and 0.189 nm after 10 ns.

For an initial density of $10^8$ cm$^{-3}$ the average size of particles is larger than for a density of $10^7$ cm$^{-3}$ after 1 s since the enlarged particle density favors the growth of sulphuric acid clusters. The average size after 1 s and after 10 s is 0.161 nm. Although the average growth is not significant between 1 s and 10 s, some clusters tend to form larger clusters. The cluster





a) $n = 10^7$ cm$^{-3}$, $T = 300$ K, $t = 1$ s

b) $n = 10^7$ cm$^{-3}$, $T = 300$ K, $t = 10$ s

c) $n = 10^7$ cm$^{-3}$, $T = 200$ K, $t = 1$ s

d) $n = 10^7$ cm$^{-3}$, $T = 200$ K, $t = 10$ s

e) $n = 10^8$ cm$^{-3}$, $T = 300$ K, $t = 1$ s

f) $n = 10^8$ cm$^{-3}$, $T = 300$ K, $t = 10$ s

**Figure 3.** The spatial distribution of all particles after 1 s (left column) and after 10 s (right column) for different densities and different temperatures projected onto the $xy$ plane. For better visibility the size of each circle is the real size of each particle multiplied by a factor of $2 \cdot 10^6$. The arrow in panel f) indicates the largest cluster of all simulations after 10 s.



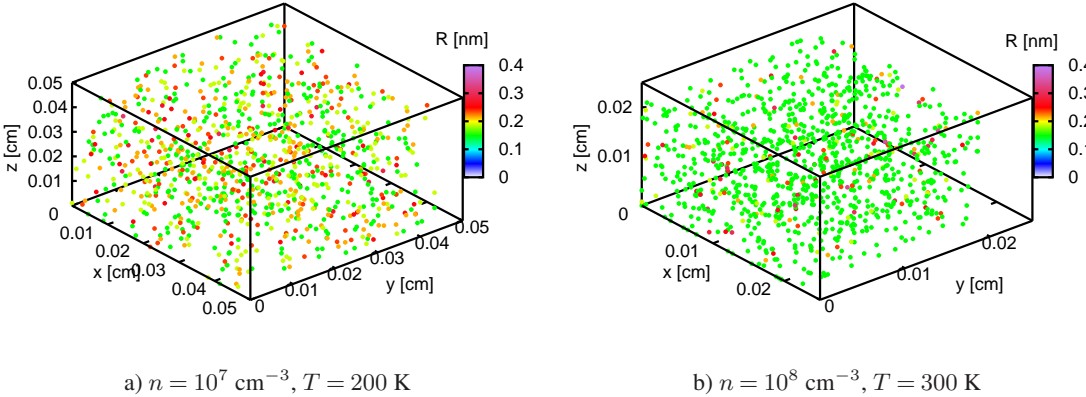

<p style="text-align:center">a) $n = 10^7$ cm$^{-3}$, $T = 200$ K          b) $n = 10^8$ cm$^{-3}$, $T = 300$ K</p>

**Figure 4.** The position of all particles in the three dimensional simulation domain after 10 s for the same densities and temperatures as in Fig. 3 d) and f). The particle size is color coded.

located at ($x = 0.01$ cm, $y = 0.02$ cm, $z = 0.02$ cm) with a radius of 0.386 nm (see arrow in panel f) is the largest cluster of all presented panels.

## 3.2 Size distribution

Fig. 5 shows the size distributions after 1 ns and after 10 ns for the same densities and temperatures as in Fig. 3. The right
5   $y$-axis in the right column displays the difference between the particle numbers after 10 s and after 1 s.

For all considered cases, the number of monomers with a size of 0.15 nm is dominant after 1 s whilst there are only a few clusters with larger sizes. For $n = 10^7$ cm$^{-3}$ and $T = 300$ K (a), there are only 14 particles larger than 0.15 nm; for $n = 10^8$ cm$^{-3}$ and $T = 300$ K (e), there are 145 particles larger than 0.15 nm and at a temperature of 200 K (c), there are 151 particles larger than 0.15 nm. For 200 K, the large number of polymers is an effect of the negligible evaporation whereas for $n = 10^8$
10   cm$^{-3}$ nucleation is driven because of the enlarged density. As we have observed in Fig. 3 c) and e), the largest particles are present for high densities instead of low temperatures.

After 10 s, the size distributions at 300 K (b,f) have not changed significantly. The right $y$-axis indicates that there is only a slight increase of particles with a size of approximately 0.35 nm. Panel d), however, shows that for a temperature of 200 K, the number of monomers has decreased enormously since there is only nucleation, but no evaporation. The right $y$-axis of panel d)
15   shows that there is an increase of polymers with a size of 0.22 nm, but no significant growth of polymers of above 0.3 nm since the diffusion coefficient $D(R)$ decreases with size and as such the duration of larger clusters until encountering neighboring particles is extended.







**Figure 5.** The size distribution of all particles after 1 s (left column) and after 10 s (right column) for the same conditions as in Fig. 3. The right $y$-axis in the right column shows the difference between the particle numbers after 10 s and after 1 s.



| | $\nu_{sim}$ [cm$^{-3}$ s$^{-1}$] | $\nu_{par}$ [cm$^{-3}$ s$^{-1}$] |
|---|---|---|
| $n = 10^7$ cm$^{-3}$, $T = 300$ K | $10^{-10.56\pm6.65}$ | $10^{-8.19}$ |
| $n = 10^7$ cm$^{-3}$, $T = 200$ K | $\approx 10^1$ | $10^{2.45}$ |
| $n = 10^8$ cm$^{-3}$, $T = 300$ K | $10^{0.34\pm6.89}$ | $10^{-4.57}$ |

**Table 1.** The nucleation rates $\nu_{sim}$ calculated from our simulations and the nucleation rates $\nu_{par}$ (Dunne et al. , 2016). For 300 K, $\nu_{sim}$ is the mean nucleation rate (Eq. 10); for 200 K $\nu_{sim}$ is the asymptotic nucleation rate.

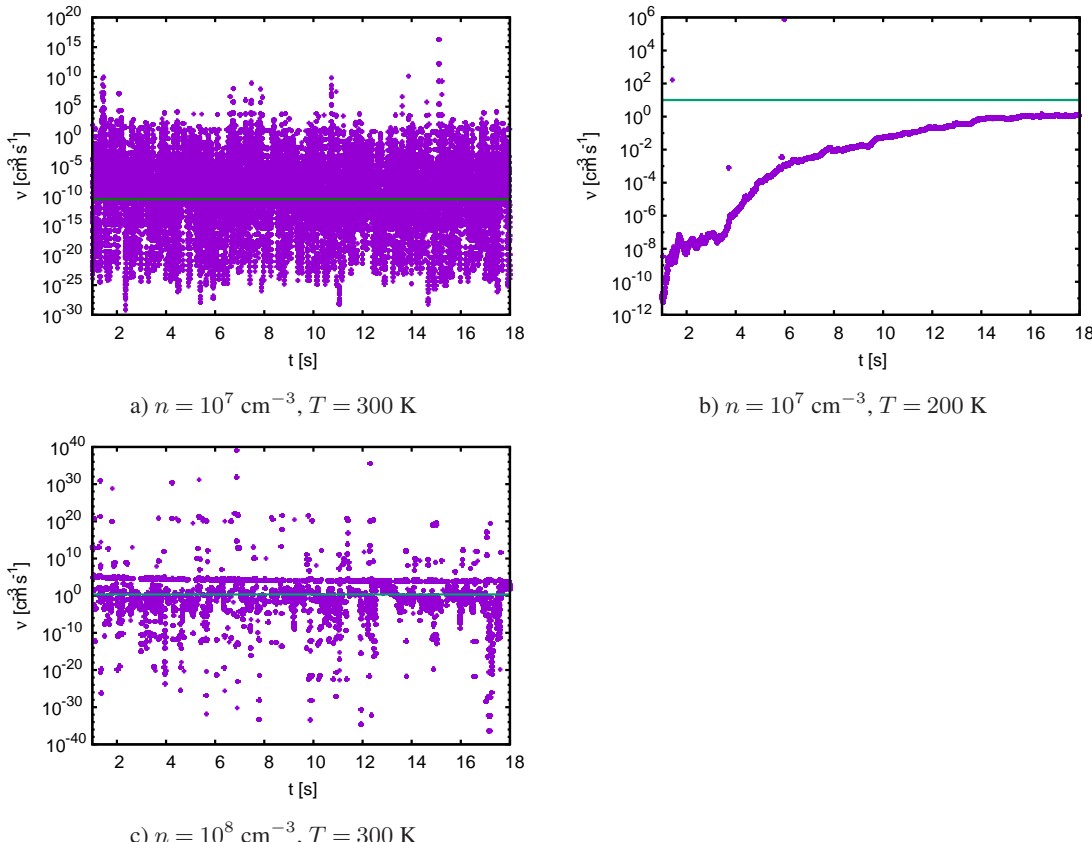

a) $n = 10^7$ cm$^{-3}$, $T = 300$ K

b) $n = 10^7$ cm$^{-3}$, $T = 200$ K

c) $n = 10^8$ cm$^{-3}$, $T = 300$ K

**Figure 6.** The nucleation rate $\nu(t)$ as a function of time for different initial densities $n$ of sulphuric acid for 200 K and for 300 K. The green line shows the calculated nucleation rate $\nu_{sim}$ (see text).





### 3.3 Nucleation rate

As Fig. 3 and 5 show, the particle sizes in all considered simulations are smaller than 0.4 nm after 10 s which complicates the determination of the nucleation rate of particles with radii above 0.4 nm. After every time step $t$ we therefore fit the size distribution, i.e. the number $N(R,t)$ of particles with size $R$, to the exponential

$$N(R,t) = a(t) \cdot e^{b(t) \cdot R}. \tag{8}$$

Subsequently the nucleation rate of particles with size $R$ as a function of time $t$ is given through

$$\nu(R,t) = \frac{N(R,t)}{V \cdot t} \tag{9}$$

where $V$ is the volume of the simulation domain.

Fig 6 shows the nucleation rate $\nu(R = 0.85 \text{ nm}, t)$ of particles with a radius of 0.85 nm as a function of time for all three
simulations. Panels a) and c) show that the nucleation rate heavily oscillates because of the continuous growth and evaporation of clusters at 300 K. For 200 K, evaporation is negligible and thus particles keep growing and consequently the nucleation rate increases as a function of time. However, the slope of the nucleation rate flattens because of the decreased diffusion coefficient for larger clusters which diffuse more slowly than small clusters and single molecules and consequently collide less with surronding particles. The size depending diffusion and the subsequent slow motion is also the reason why there is no
significant difference between Fig. 3 c) and d). Whereas for 300 K the evaporation impedes the formation of large clusters, the smaller diffusion coefficient at 200 K and its size dependence reduce the probability of a large particle colliding with any other.

The green line in panels a) and c) of Fig. 6 shows the mean nucleation rate

$$\langle \nu \rangle = \frac{1}{\mathcal{T}} \sum_t \nu(0.85 \text{ nm}, t) \tag{10}$$

of particles with a radius of 0.85 nm within time interval $\mathcal{T}$. Dunne et al. (2016) simulated the formation of atmospheric aerosol
particles in extensive laboratory experiments. They determined the nucleation rates for aerosols at different temperatures and for different compounds and presented parametrisations $\nu_{par}$ for the nucleation rate of clusters with a radius of 0.85 nm as a function of density and temperature.

Table 1 compares the nucleation rates $\nu_{sim}$ calculated from our simulations with the nucleation rates $\nu_{par}$ obtained by Dunne et al. (2016). For 300 K, $\nu_{sim}$ is the mean nucleation rate (10) whereas for 200 K $\nu_{sim}$ is the asymptotic value of
approximately $10^1$ cm$^{-3}$ s$^{-1}$ (green line in Fig. 6 b). Note that for longer runtimes the mean nucleation for 200 K will become comparable to the asymptotic nucleation rate since $\nu(0.85 \text{ nm}, t)$ is increasing monotonously in time. In all cases $\nu_{sim}$ is comparable to the values obtained by Dunne et al. (2016), hence there is a good agreement within the error bars.

### 4 Conclusions

We have presented a particle Monte Carlo model tracing individual $H_2SO_4$ molecules and clusters for different initial densities,
$n$, of sulphuric acid molecules and for different temperatures $T$ taking the growth by collision of particles and evaporation by single $H_2SO_4$ molecules into account.




Three cases were considered, $n = 10^7$ cm$^{-3}$ with temperature either $T = 300$ K or $T = 200$ K, and one case with $n = 10^8$ cm$^{-3}$ and $T = 300$ K. In the two cases with the low sulphuric acid concentration, there is no significant growth between 1 s and 10 s, in accordance with a low nucleation rate. However, for the higher sulphuric acid concentration, growth of clusters between 1 s and 10 s is observed.

The simulations make it possible to calculate the size distribution based on individual particle sizes. For 300 K, most particles are single molecules and only few particles exist with sizes larger than 0.2 nm. This behaviour is rather independent of the initial density. In contrast at 200 K, evaporation becomes negligible and thus the size distribution consists of fewer monomers compared to the case with $T = 300$ K which increases the number of clusters above 0.2 nm. No significant growth is observed for clusters above 0.22 nm for $n = 10^7$ cm$^{-3}$ and $T = 200$ K since the diffusion coefficient for 200 K is smaller than for 300

K and decreases with increasing cluster size. The largest growth and thereby clusters are found in the case for $n = 10^8$ cm$^{-3}$ and $T = 300$ K.

   Finally we have calculated the nucleation rates as a function of time and the mean nucleation rates averaged over time. Since for 300 K there is an interplay between nucleation and evaporation, the nucleation rate oscillates around the mean nucleation rate. For 200 K, evaporation is negligible and as such the nucleation rate increases with time. However, it tends to an upper limit

since the probability of two particles colliding with each other is reduced as an effect of the decreased diffusion coefficient.

   We compared the nucleation rates with values experimentally obtained by Dunne et al. (2016). Within the given error bars we see a good agreement between our simulation results and experimental values which serves as a benchmark for our Monte Carlo code. We therefore conclude that the physics implemented in the present Monte Carlo model is appropriate to simulate the growth of sulphuric acid clusters. Its main advantage is that it traces individual particles and therefore reflects their distinct

nature and as such reality much better than for example kinetic numerical models or pure parametrisations.

   In a forthcoming paper, we will present a more sophisticated model where we include HSO$_4^-$ ions and investigate their influence on the nucleation rate.

*Acknowledgements.* The research was partly funded by the Marie Curie Actions of the European Union's Seventh Framework Programme (FP7/2007-2013) under REA grant agreement n$^o$ 609405 (COFUNDPostdocDTU).



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
