# Peer review of "A 3D particle Monte Carlo approach to studying nucleation"

_Atmospheric Chemistry and Physics, 2017_

## Referee Comment (RC1) · Anonymous Referee #1 · 4 Jul 2017

Köhn, Enghoff and Svensmark present a new 3-dimensional Monte-Carlo - based model for sulfuric acid clustering. Essentially, the model combines an extremely simplified parameterization for the cluster thermodynamics (and molecule-cluster or cluster-cluster interactions) with a sophisticated approach for simultaneously tracking a large number of molecules (and clusters) in 3-dimensional space. The authors speculate that "If each molecule could be tracked in a three dimensional space it might be possible to achieve new insights into the process of nucleation". I agree that this could indeed be possible, and was eagerly expecting to see such new insights in the results-section of the paper. Sadly, I found the paper entirely lacking in new insights, as it consisted solely of a (somewhat cursory and partially inconsistent) technical description of the model, and some (not particularly convincing) benchmarking. I'm not convinced this

paper warrants publication as a stand-alone article in ACP - I would suggest the authors either submit a corrected and expanded version of technical description as a technical note somewhere, or alternatively combine it with simulations (e.g. on the proposed HSO4- clustering) that would actually provide some genuinely new insights. Regardless of what the authors decide to do, I hope they take my comments below into account:

Technical comments/questions concerning the simulation details:

1) In equation 1 the motion of particles between two time steps is governed by a random step. Yet later (e.g. in discussing equations 5 & 6) the authors talk about the velocities of particles. What meaning do the velocities have if the particle motion is based on random numbers? How do the velocities from equations 5 & 6 enter into equation 1?

2) Surely the time step should be significantly *smaller* than the average time it takes for particles to diffuse the inter-particle separation? Setting the time step equal to this seems to risk numerical issues... Especially for the high-density studies where the time step used is now almost five times the average time! Sensitivity studies should definitely be performed to establish that the results are converged with respect to the time step.

3) What is the justification of using r=0.85 nm as the threshold for a particle being "nucleated"? This seems to be quite small for such a threshold, e.g. compared to any experimental nucleation study.

4) The authors cite and apparently make use of data from Yu et al 2005, which is (as the very title suggests) a sulfuric acid - WATER nucleation model. Yet they do not mention water anywhere in the paper. They are presumably following in Yu's footsteps and using a quasi-unary approach, where the effect of water is implicitly accounted for. But then they should specify this, and also give the relative or absolute humidity the simulations are (again implicitly) performed at. They might be tempted to answer that the RH is assumed to be 0% (which incidentally would leave the results atmospherically irrelevant).

But then they can not use Dunne et al as comparison, because as that paper states: "For the purposes of our inorganic nucleation parameterization, we therefore assume that no nucleation occurs below 10% RH". So either the authors implicitly have water in their system, in which case A LOT of description on this is needed, or then they must abandon any comparison with the Dunne et al paper.

5)As an additional comment to that above, the slope of the evaporation coefficient shown in Figure 1 is curiously flat compared to the original data from Yu et al 2005, (Figure 4 in that paper), especially when considering that the y-axis in the present paper is R and that in the Yu paper is number of molecules (which increases as R^3). The actual evaporation rate numbers are also curiously low especially if the intention is to model dry (RH 0%) sulfuric acid clustering, as water is well-known to stabilize the clusters (i.e. the evaporation rate for T=300K and RH 0% can hardly be LOWER than that at R=298 K and RH 50%, as given in Fig 4 of the Yu paper) .I'm also not convinced that it is even possible to obtain a completely "dry & pure" sulfuric acid vapor pressure, but I'd like to see the authors reply to the H2O issue before expanding on this further.

Comments on the interpretation and comparison of results:

6)The authors correctly state that Monte Carlo models are well suited to capture rare events, of which nucleation certainly is an example. But are they actually capturing such rare events? In the beginning of section 3.3. the authors state that "As Fig. 3 and 5 show, the particle sizes in all considered simulations are smaller than 0.4 nm after 10 s which complicates the determination of the nucleation rate of particles with radii above 0.4 nm." I could say this more strongly: if no particles larger than 0.4 nm are formed then the nucleation rate of particles with radii larger than that is actually ZERO. So in other words the authors simulations are not actually showing nucleation (in which the particles continue growing and growing and growing after passing a critical size), just some limited initial clustering. The curve fitting of equations 8 and 9 seems to be an essentially meaningless exercise - the oscillations of the fitted rates, as well as the enormous - over 12 orders of magnitude - error bars from the fit are to me additional

indications of this. Even if the curve fitting were somehow justified (for which I would like to see MUCH more justifications) the approach in my opinion destroys the original advantage of the MC method: instead of actually observing rare events, the authors are now back to the same sort of indirect fitting-based approach that plagues many other model types. Comparing to the Dunne et al parameterizations, the lack of actual nucleation observations is not actually very surprising: the simulation box size in the low-density (1E7 particles per cm^3) simulations is, if I understand correctly, 0.000125 cm^3, so with a run time of 10 s the nucleation rate would need to be on the order of 800 new particles per cm3 and s to (on average) see one nucleation event per simulation. I would suggest the authors redo their benchmarking simulations with conditions that should (e.g. based on the Dunne et al parameterization) actually lead to mutiple events per simulation, and then re-analyze their results using actual particle formation rates, not indirect fits - then there would actually be some genuine advantages to using a Monte-Carlo model.

7)Details should be given on exactly how the Dunne et al comparison numbers were obtained, this is currently not at all clear.

8)The authors claim thaty "the implemented physics is sufficient enough to study the nucleation of neutral sulphuric acid clusters in order to benchmark our model". This may or may not be true, but given the issues described above, and the 13 orders of magnitude uncertainty in the fits even according to the authors own estimate, the claim is far from proven.

---

## Referee Comment (RC2) · Anonymous Referee #2 · 4 Jul 2017

The authors present a Monte Carlo scheme for simulating the evolution of a set of particles in a spatial volume, designed to represent sulphuric acid monomers and their clusters. The particles combine together when a stochastic spatial move takes them within an overlap distance of one another, and they fragment with the emission of a monomer according to a stochastic thermal evaporation process.

In my opinion, there are numerous problems with this manuscript and I cannot recommend publication.

The authors state in the introduction that the technique might possibly provide 'new insights into the process of nucleation'. However, I am not persuaded that this has been achieved, and the authors do not enlarge on this claim in the conclusions. The MC method presented is essentially a numerical implementation of the stochastic evolution

of a cluster size distribution, and I suspect that the outcomes are consistent with much less elaborate treatments, such as a simple Becker-Doering rate equation approach. Namely, the events are infrequent in the situations considered (and probably do not give rise to spatial correlations between potential collision partners) and are dominated by single monomer attachment and detachment processes rather than cluster-cluster collisions. If the reported cluster size distributions were compared with the results of such a rate equation scheme, involving the solution of a few ODEs for a set of spatially averaged cluster populations, I would expect to find the outcomes compatible. This would have been a very useful benchmarking procedure to assess the value of using a 3D approach.

The updating of the configuration is not entirely clear. In a couple of places, the authors refer to the velocities of the particles. However, the dynamics, in equation (1), seems to involve stochastic moves in spatial position only. What is the purpose of considering the velocities? Is the update a combination of ballistic motion and diffusive jumps? The discussion seem to imply only diffusion.

Furthermore, the selection of the direction of diffusive motion does not seem right. The spherical polar angles theta and phi are sampled uniformly from their respective ranges (page 3, line 10). This is fine for phi, but the angle theta should be sampled nonuniformly (in proportion to sin(theta)) in order to provide a uniform sampling of the 4 pi of solid angle. Uniform sampling of theta improperly favours the polar regions of a globe rather than the equator and if this is indeed the scheme implemented, then the particles will tend to diffuse up and down (with respect to the z axis of the simulation cell) and not sideways. Since the MC scheme is intended to be physically representative, this is a significant drawback. But perhaps this is a typo in the manuscript.

The authors are apparently unaware that other 3D 'brute force' numerical simulations have previously been carried out to investigate the process of nucleation. These have involved molecular dynamics rather than (kinetic) Monte Carlo configurational evolution, but the aim is very similar, as is the computational expense. Recent papers (2014-

16) by Diemand, Angelil and the Tanakas should be consulted, and older studies by Wedekind et al (2006-2007).

The authors seek to compare their results with experimental values of nucleation rates for sulphuric acid clusters of radius 0.85 nm, even though no clusters of this size are generated in their simulations. In order to make this assessment, they fit an exponential function (equation (8)) to the cluster size distribution at each timestep and extrapolate to the target size. This is not a convincing approach since it is likely to introduce large uncertainty. But even if there were reason to use such a function, the nucleation rate is not a mean population at a certain size divided by the elapsed time, as given in equation (9); instead it is a current or flow of clusters through that size. In an asymptotic stationary state, the definition used by the authors would produce a nucleation rate of zero. Only if there were a maximum size cluster in the scheme, receiving clusters from the size below but not losing them to sizes above or below (i.e. equivalent to a cluster sink), would such a definition be appropriate. The extracted nucleation rates are stated to be compatible with experimental values of sulphuric acid, within error bars, but the statistical errors quoted in Table 1 are very large, and the claim of consistency is weak.

Finally, I have reservations about the assumed evaporation rate in equation (7). The authors have introduced a factor of $R_1/R$ on the grounds that the saturated vapour pressure over a curved interface is lower than that over a flat interface, such that the evaporation rate should increase as the radius decreases. But it is the Kelvin term (the exponential in equation (7)) that represents this effect: I do not see why a new factor is required. The Kelvin term is a consequence of the capillarity approximation used in the construction of the excess free energy of cluster formation from monomers, and so the extra factor corresponds to a non-classical term (in the sense of an addition to Classical Nucleation Theory (CNT)) in the excess free energy. The introduction of such a new term has not been justified.

In passing, setting the evaporation rate equal to the Kelvin form ought to lead to a nucleation rate consistent with CNT, at least in some circumstances, and this would also

be a useful benchmark exercise. Another useful benchmark would be to compare the simulations at 200 K with known analytical results for a Brownian coagulation model.

---

## Referee Comment (RC3) · Anonymous Referee #3 · 10 Jul 2017

Review on the manuscript "A 3D particle Monte Carlo approach to studying nucleation" by Christoph Köhn, Martin Bødker Enghoff, and Henrik Svensmark.

The present manuscript deals with simulation on growth of sulphuric acid clusters. The authors call the modelling method Monte Carlo. I would avoid using such name in this context. Technically, method employs random number generator, so it can be called Monte Carlo method. However it does not give any hint on details of modeling. Furthermore, in molecular physics the term Monte Carlo simulations is reserved for the group of methods modelling equilibrium distribution functions of particular variables. In the manuscript such examples are given: Kusaka et al.(1998), and Kathmann and Hale (2001). I would call the employed method: simulation of random walk governed by diffusion equation. This remark is just a suggestion and has no influence on the

referee's decision.

There is some confusion in description of the method. It is mentioned in the abstract and Eq. (6) says that the velocity of the merged particle is defined by conservation of momentum. On the other hand, according to the description of the method the initial velocities are not assigned to the molecules; neither positions nor the probability of sticking depend on the velocities. The particle positions are completely defined by diffusion equations and the velocity is just unnecessary detail in the simulations.

It is not clearly explained how the evaporation is incorporated into modeling. The correct procedure is compare the value exp(-$\gamma$ $\Delta$t) to the generated random number r between 0 and 1. If exp(-$\gamma$ $\Delta$t) > r, the evaporation doesn't happen, otherwise – yes. Perhaps, it is done so but just not said.

As far as I could deduce from the text all values are obtained from one single run (one realization of the random process). The spatial and size distributions and size from one single run are not comparable to experiments. It is necessary to have big number of realizations (I recommend 10 000 - 100 000 for smooth results) with different initial positions of molecules. The procedure for the calculating of the average nucleation rate is correct since the ensemble averaging can be substituted with the time averaging.

The major flaw in the manuscript is that movement of particles is considered as diffusional. It is well known that if ratio L/R (Knudsen number) much more than 1, the collision frequency between the particles is defined by the rate from the gas kinetic theory rather than by the diffusional rate constant; here L is the mean free path, R is the size of the particle. Despite that most of the way particles move in the diffusional regime at large Knudsen numbers the limiting stage for collision is the last step when the particles move in the free-molecular regime. Fuchs ("Mechanics of Aerosols", Pergamon Press, London, 1964) discusses this problem at length in the book. For particles as large as 0.85 nm, the pressure of 1 bar, and temperatures 200-300 the Knudsen number is roughly 40 – 60.

The method employed in this study is not novel. If it were applied properly, it should give results identical to the ones from the solution of Smoluchowski problem since no new physics is introduce into the model.

I do not recommend the manuscript for publication in the journal Atmospheric Chemistry and Physics.
* * *

---

## Referee Comment (RC4) · Anonymous Referee #4 · 31 Jul 2017

This paper attempts a Monte-Carlo study of the vapor-phase evolution of sulfuric acid molecules under clustering and evaporation of monomer. These species are treated in the paper as spheres with initial monomer size of 0.15nm and nucleation rate derived from the formation of clusters at 0.85nm. It is not clear what the critical nucleation cluster size is, but it is presumably smaller than 0.85nm. Initial gas-phase concentrations are at $10^7$ and $10^8$ sulfuric acid molecules per cc in an ambient atmosphere of pressure 1 bar.

Under the simulated conditions the mean-free path is about 70nm, which is much larger than even the 0.85nm size considered for nucleation to occur. Therefore clusters are in the free-molecular size regime (mean free path much larger than the particle size). The diffusion controlled regime occurs when the mean free path is much smaller than

the particle size. So it is not clear why the authors use the diffusion-controlled model of Fransaer and Penner, cited on page 12 line 14, which applies to Brownian-like diffusive motion of individual molecules or ions in dilute aqueous electrolytes. Unfortunately, this is a problem with the manuscript that affects the growth law throughout and makes it unacceptable for publication.

Additional comments: The authors appear to be dealing with a much simpler problem than they treat: The clusters are treated as spheres and there is no explicit structure or inter-molecular potential assigned to them. The authors might as well be treating clusters in the capillarity approximation of nucleation theory, which treats clusters as spheres with sharp interface and no inherent structure other than that of the bulk liquid (for density) and a surface tension for evaporation rate from the Kelvin relation that is much like the author's Eq. 7. Such simplicity suggests that collisions might be tracked through Fokker-Planck type diffusion-drift coagulation/evaporation equations without knowing their precise positions in 3D – as only a detailed dynamics would require. Then MC could be used for calculation of arrivals and departures based statistically on cluster number densities and evaporation rates. Presumably the authors would end up with a Boltzmann distribution of cluster size. The reviewer is not an expert in molecular simulations but it does seem that detailed knowledge of 3D molecule/cluster position is overkill for spherically symmetric molecule/cluster interaction. Moreover, such detailed knowledge might be washed out anyway given the assumption of randomness that is already part of the MC method.

---

## Author Comment (AC1) · 22 Sep 2017

FMs. Ref. No.: ACP-2017-269

Title: A 3D particle Monte Carlo approach to studying nucleation

*We thank the referees for their careful reading and useful suggestions. Our answers are inserted into the reports in italics.*

*Furthermore, we have clarified a few issues in the paper, added additional material and improved language and structure of the paper.*

**Reviewer #1**:

Köhn, Enghoff and Svensmark present a new 3-dimensional Monte-Carlo – based model for sulfuric acid clustering. Essentially, the model combines an extremely simplified parameterization for the cluster thermodynamics (and molecule-cluster or clustercluster interactions) with a sophisticated approach for simultaneously tracking a large number of molecules (and clusters) in 3-dimensional space. The authors speculate that "If each molecule could be tracked in a three dimensional space it might be possible to achieve new insights into the process of nucleation". I agree that this could indeed be possible, and was eagerly expecting to see such new insights in the results-section of the paper. Sadly, I found the paper entirely lacking in new insights, as it consisted solely of a (somewhat cursory and partially inconsistent) technical description of the model, and some (not particularly convincing) benchmarking. I'm not convinced this paper warrants publication as a stand-alone article in ACP - I would suggest the authors either submit a corrected and expanded version of technical description as a technical note somewhere, or alternatively combine it with simulations (e.g. on the proposed HSO4- clustering) that would actually provide some genuinely new insights.

*We thank the referee for this estimation. Indeed, we agree that the current manuscript is a rather technical one and we have decided to withdraw it from ACP and submit it to Journal of Computational Physics.*

Regardless of what the authors decide to do, I hope they take my comments below into account:

Technical comments/questions concerning the simulation details:

1)In equation 1 the motion of particles between two time steps is governed by a random step. Yet later (e.g. in discussing equations 5 & 6) the authors talk about the velocities of particles. What meaning do the velocities have if the particle motion is based on random numbers? How do the velocities from equations 5 & 6 enter into equation 1?

*The referee is right. Although we have mentioned the velocities in the paper, we did not include them into our computational model. We have deleted the corresponding passages.*

2)Surely the time step should be significantly *smaller* than the average time it takes for particles to diffuse the inter-particle separation? Setting the time step equal to this seems to risk numerical issues... Especially for the high-density studies where the time step used is now almost five times the average time! Sensitivity studies should definitely be performed to establish that the results are converged with respect to the time step.

*The referee is right that the time step should not be too large to risk numerical issues. We have now chosen smaller time steps than $1/(2D_0 \, n^{2/3})$. We have specified this further in the paper.*

3)What is the justification of using r=0.85 nm as the threshold for a particle being "nucleated"? This seems to be quite small for such a threshold, e.g. compared to any experimental nucleation study.

*We have used this value in order to compare our nucleation rates with the ones by [Dunne et al., 2016. Global atmospheric particle formation from CERN CLOUD measurements. Science, vol. 354, pp. 1119–1124]. We have clarified this in the paper.*

4)The authors cite and apparently make use of data from Yu et al 2005, which is (as the very title suggests) a sulfuric acid - WATER nucleation model. Yet they do not mention water anywhere in the paper. They are presumably following in Yu's footsteps and using a quasi-unary approach, where the effect of water is implicitly accounted for. But then they should specify this, and also give the relative or absolute humidity the simulations are (again implicitly) performed at. They might be tempted to answer that the RH is assumed to be 0% (which incidentally would leave the results atmospherically irrelevant). But then they can not use Dunne et al as comparison, because as that paper states: "For the purposes of our inorganic nucleation parameterization, we therefore assume that no nucleation occurs below 10% RH". So either the authors implicitly have water in their system, in which case A LOT of description on this is needed, or then they must abandon any comparison with the Dunne et al paper.

*As in the paper by Yu et al., we have updated the model to include the water content implicitly through the molar fraction defined in the same paper. We have chosen a relative humidity of 50%. We have added all necessary information about how we have implemented the relative humidity implicitly.*

5)As an additional comment to that above, the slope of the evaporation coefficient shown in Figure 1 is curiously flat compared to the original data from Yu et al 2005, (Figure 4 in that paper), especially when considering that the y-axis in the present paper is R and that in the Yu paper is number of molecules (which increases as R^3). The actual evaporation rate numbers are also curiously low especially if the intention is to model dry (RH 0%) sulfuric acid clustering, as water is well-known to stabilize the clusters (i.e. the evaporation rate for T=300K and RH 0% can hardly be LOWER than that at R=298 K and RH 50%, as given in Fig 4 of the Yu paper) .I'm also not convinced that it is even possible to obtain a completely "dry & pure" sulfuric acid vapor pressure, but I'd like to see the authors reply to the H2O issue before expanding on this further.

*We have revised the calculation of the evaporation coefficient for a relative humidity of 50% and for different temperatures. We now made the mass, the radius, the surface tension of all particles as well as the concentration of sulphuric acid molecules vapor molecules a function of the mole*

*fraction.*

Comments on the interpretation and comparison of results:

6)The authors correctly state that Monte Carlo models are well suited to capture rare events, of which nucleation certainly is an example. But are they actually capturing such rare events? In the beginning of section 3.3. the authors state that "As Fig. 3 and 5 show, the particle sizes in all considered simulations are smaller than 0.4 nm after 10 s which complicates the determination of the nucleation rate of particles with radii above 0.4 nm." I could say this more strongly: if no particles larger than 0.4 nm are formed then the nucleation rate of particles with radii larger than that is actually ZERO. So in other words the authors simulations are not actually showing nucleation (in which the particles continue growing and growing and growing after passing a critical size), just some limited initial clustering. The curve fitting of equations 8 and 9 seems to be an essentially meaningless exercise - the oscillations of the fitted rates, as well as the enormous - over 12 orders of magnitude - error bars from the fit are to me additional indications of this. Even if the curve fitting were somehow justified (for which I would like to see MUCH more justifications) the approach in my opinion destroys the original advantage of the MC method: instead of actually observing rare events, the authors are now back to the same sort of indirect fitting-based approach that plagues many other model types. Comparing to the Dunne et al parameterizations, the lack of actual nucleation observations is not actually very surprising: the simulation box size in the low-density (1E7 particles per cm^3) simulations is, if I understand correctly, 0.000125 cm^3, so with a run time of 10 s the nucleation rate would need to be on the order of 800 new particles per cm3 and s to (on average) see one nucleation event per simulation. I would suggest the authors redo their benchmarking simulations with conditions that should (e.g. based on the Dunne et al parameterization) actually lead to mutiple events per simulation, and then re-analyze their results using actual particle formation rates, not indirect fits - then there would actually be some genuine advantages to using a Monte-Carlo model.

*With the updated model on the relative humidity and the corrected evaporation coefficients, we have not seen any large clusters for a temperature of 300 K, as expected. However, for temperatures of 200 K (see Figure 1 of the reply) and 238 K, there is a substantial number of clusters above 0.85 nm. We will include this into the new version of the paper. Hence, we now also calculate the nucleation rate  for clusters above 0.85 nm.*

7)Details should be given on exactly how the Dunne et al comparison numbers were obtained, this is currently not at all clear.

*This is indeed a very good suggestion. We have added the parametrizations used by Dunne et al. into the manuscript.*

8)The authors claim that "the implemented physics is sufficient enough to study the nucleation of neutral sulphuric acid clusters in order to benchmark our model". This may or may not be true, but given the issues described above, and the 13 orders of magnitude uncertainty in the fits even according to the authors own estimate, the claim is far from proven.

*Actually, since we now observe nucleation for 200 K and 238 K, we calculate the nucleation rate by*

*actually counting clusters above 0.85 nm. This minimizes the many orders of magnitude uncertainty; for 300 K we conclude to have a nucleation rate of 0.*

*Figure 1: The size distribution of all particles after 50 s for n=$10^7$ cm$^{-3}$ and T = 200 K.*

[Figure]

---

## Author Comment (AC2) · 22 Sep 2017

FMs. Ref. No.: ACP-2017-269

Title: A 3D particle Monte Carlo approach to studying nucleation

*We thank the referees for their careful reading and useful suggestions. Our answers are inserted into the reports in italics.*

*Furthermore, we have clarified a few issues in the paper, added additional material and improved language and structure of the paper.*

**Reviewer #2**:

The authors present a Monte Carlo scheme for simulating the evolution of a set of particles in a spatial volume, designed to represent sulphuric acid monomers and their clusters. The particles combine together when a stochastic spatial move takes them within an overlap distance of one another, and they fragment with the emission of a monomer according to a stochastic thermal evaporation process. In my opinion, there are numerous problems with this manuscript and I cannot recommend publication.

The authors state in the introduction that the technique might possibly provide 'new insights into the process of nucleation'. However, I am not persuaded that this has been achieved, and the authors do not enlarge on this claim in the conclusions. The MC method presented is essentially a numerical implementation of the stochastic evolution of a cluster size distribution, and I suspect that the outcomes are consistent with much less elaborate treatments, such as a simple Becker-Doering rate equation approach.

*So far what we get out of the model is indeed nucleation rates and size distributions which can be obtained faster using other methods, as the referee mentions. We do this in an attempt to validate the model. If we are convinced that the model produced reliable results regarding the parameters where we can compare to other peoples work then we can, with some confidence, start looking for any new information that can be gained from seeing the entire 3D-picture. We have added this now to the conclusion and outlook section.*

Namely, the events are infrequent in the situations considered (and probably do not give rise to spatial correlations between potential collision partners) and are dominated by single monomer attachment and detachment processes rather than cluster-cluster collisions. If the reported cluster size distributions were compared with the results of such a rate equation scheme, involving the solution of a few ODEs for a set of spatially averaged cluster populations, I would expect to find the outcomes compatible. This would have been a very useful benchmarking procedure to assess the value of using a 3D approach.

*We agree that this would have been a possible approach. However, we think that the comparison*

*with Dunne et al. is a valid approach, too.*

The updating of the configuration is not entirely clear. In a couple of places, the authors refer to the velocities of the particles. However, the dynamics, in equation (1), seems to involve stochastic moves in spatial position only. What is the purpose of considering the velocities? Is the update a combination of ballistic motion and diffusive jumps? The discussion seem to imply only diffusion.

*The referee is right. Although we have mentioned the velocities in the paper, we did not include them into our computational model. We have deleted the corresponding passages.*

Furthermore, the selection of the direction of diffusive motion does not seem right. The spherical polar angles theta and phi are sampled uniformly from their respective ranges (page 3, line 10). This is fine for phi, but the angle theta should be sampled nonuniformly (in proportion to sin(theta)) in order to provide a uniform sampling of the 4 pi of solid angle. Uniform sampling of theta improperly favours the polar regions of a globe rather than the equator and if this is indeed the scheme implemented, then the particles will tend to diffuse up and down (with respect to the z axisof the simulation cell) and not sideways. Since the MC scheme is intended to be physically representative, this is a significant drawback. But perhaps this is a typo in the manuscript.

*Yes, the referee is right. It should have been $\vartheta = arccos(2r_3-1)$. We have corrected for this in our manuscript as well as in our code.*

The authors are apparently unaware that other 3D 'brute force' numerical simulations have previously been carried out to investigate the process of nucleation. These have involved molecular dynamics rather than (kinetic) Monte Carlo configurational evolution, but the aim is very similar, as is the computational expense. Recent papers (2014-16) by Diemand, Angelil and the Tanakas should be consulted, and older studies by Wedekind et al (2006-2007).

*We do not disagree that there have been previous papers about molecular dynamics. However, so far there has not been any molecular dynamics approach towards the nucleation of sulphuric acid clusters. We mention now papers [J. Diemand et al., 2013. Large scale molecular dynamics simulations of homogeneous nucleation. J. Chem. Phys., vol. 139, 074309. R. Angelil et al., 2015. Homogeneous SPC/E water nucleation in large molecular dynamics simulations. J. Chem. Phys., vol. 143, 064507] in the introduction.*

The authors seek to compare their results with experimental values of nucleation rates for sulphuric acid clusters of radius 0.85 nm, even though no clusters of this size are generated in their simulations. In order to make this assessment, they fit an exponential function (equation (8)) to the cluster size distribution at each timestep and extrapolate to the target size. This is not a convincing approach since it is likely to introduce large uncertainty. But even if there were reason to use such a function, the nucleation rate is not a mean population at a certain size divided by the elapsed time, as given in equation (9); instead it is a current or flow of clusters through that size. In an asymptotic stationary state, the definition used by the authors would produce a nucleation rate of zero. Only if there were a maximum size cluster in the scheme, receiving clusters from the size below but not losing them to sizes above or below (i.e. equivalent to a cluster sink), would such a definition be appropriate. The extracted nucleation rates are stated to be compatible with experimental values of

sulphuric acid, within error bars, but the statistical errors quoted in Table 1 are very large, and the claim of consistency is weak.

*With the corrected model we now observe clusters with sizes above 0.85 nm. Figure 1 of this reply shows the size distribution after 50 s for a temperature of 200 K. We now calculate the nucleation rate based on counting the actual particle number.*

Finally, I have reservations about the assumed evaporation rate in equation (7). The authors have introduced a factor of R_1/R on the grounds that the saturated vapour pressure over a curved interface is lower than that over a flat interface, such that the evaporation rate should increase as the radius decreases. But it is the Kelvin term (the exponential in equation (7)) that represents this effect: I do not see why a new factor is required. The Kelvin term is a consequence of the capillarity approximation used in the construction of the excess free energy of cluster formation from monomers, and so the extra factor corresponds to a non-classical term (in the sense of an addition to Classical Nucleation Theory (CNT)) in the excess free energy. The introduction of such a new term has not been justified.

*Thank you for making us aware that the size dependency is included in the exponential. We have revised the whole calculation of the evaporation frequency. We have added these changes in the paper.*

In passing, setting the evaporation rate equal to the Kelvin form ought to lead to a nucleation rate consistent with CNT, at least in some circumstances, and this would also be a useful benchmark exercise. Another useful benchmark would be to compare the simulations at 200 K with known analytical results for a Brownian coagulation model.

*An analytical benchmark would certainly also be useful, but has not been included at this stage.*

*Figure 1: The size distribution of all particles after 50 s for n=10$^7$ cm$^{-3}$ and T = 200 K.*

[Figure]

---

## Author Comment (AC3) · 22 Sep 2017

FMs. Ref. No.: ACP-2017-269

Title: A 3D particle Monte Carlo approach to studying nucleation

*We thank the referees for their careful reading and useful suggestions. Our answers are inserted into the reports in italics.*

*Furthermore, we have clarified a few issues in the paper, added additional material and improved language and structure of the paper.*

**Reviewer #3**:

Review on the manuscript "A 3D particle Monte Carlo approach to studying nucleation" by Christoph Köhn, Martin Bødker Enghoff, and Henrik Svensmark. The present manuscript deals with simulation on growth of sulphuric acid clusters. The authors call the modelling method Monte Carlo. I would avoid using such name in this context. Technically, method employs random number generator, so it can be called Monte Carlo method. However it does not give any hint on details of modeling. Furthermore, in molecular physics the term Monte Carlo simulations is reserved for the group of methods modelling equilibrium distribution functions of particular variables. In the manuscript such examples are given: Kusaka et al.(1998), and Kathmann and Hale (2001). I would call the employed method: simulation of random walk governed by diffusion equation. This remark is just a suggestion and has no influence on the referee's decision.

*We thank the referee for this suggestion. However, in other scientific areas, Monte Carlo codes refer to particle codes involving the random walk of electrons. One example, also mentioned in the introduction, is the motion of electrons in streamer discharges, e.g. in [C. Köhn and U. Ebert, 2014. The structure of ionizations showers in air generated by electrons with 1 MeV energy or less. Plasma Sour. Sci. Technol., vol. 23, 045001]. Hence, we have decided to keep the title as it is.*

There is some confusion in description of the method. It is mentioned in the abstract and Eq. (6) says that the velocity of the merged particle is defined by conservation of momentum. On the other hand, according to the description of the method the initial velocities are not assigned to the molecules; neither positions nor the probability of sticking depend on the velocities. The particle positions are completely defined by diffusion equations and the velocity is just unnecessary detail in the simulations.

*The referee is right. Although we have mentioned the velocities in the paper, we did not include them into our computational model. We have deleted the corresponding passages.*

It is not clearly explained how the evaporation is incorporated into modeling. The correct procedure is compare the value exp(- γt) to the generated random number r between 0 and 1. If exp(- γt) > r, the evaporation doesn't happen, otherwise – yes. Perhaps, it is done so but just not said.

*In order to check for evaporation we use the criterium $r < 1-exp(-\gamma t), r \in [0,1)$ which is*

*equivalent to the criterium mentioned by the referee. We have clarified this.*

As far as I could deduce from the text all values are obtained from one single run (one realization of the random process). The spatial and size distributions and size from one single run are not comparable to experiments. It is necessary to have big number of realizations (I recommend 10 000 - 100 000 for smooth results) with different initial positions of molecules. The procedure for the calculating of the average nucleation rate is correct since the ensemble averaging can be substituted with the time averaging.

*Of course, the statistics would be better if more simulation runs were performed and the average were taken. However, this is not crucial for the benchmarking and thus for our conclusions.*

The major flaw in the manuscript is that movement of particles is considered as diffusional. It is well known that if ratio L/R (Knudsen number) much more than 1, the collision frequency between the particles is defined by the rate from the gas kinetic theory rather than by the diffusional rate constant; here L is the mean free path, R is the size of the particle. Despite that most of the way particles move in the diffusional regime at large Knudsen numbers the limiting stage for collision is the last step when the particles move in the free-molecular regime. Fuchs ("Mechanics of Aerosols", Pergamon Press, London, 1964) discusses this problem at length in the book. For particles as large as 0.85 nm, the pressure of 1 bar, and temperatures 200-300 the Knudsen number is roughly 40 – 60.

*The mean free path of sulphuric acid clusters in air is approx. 10-60 nm [J.H. Seinfeld and S.N. Pandis, 2006. Atmospheric Chemistry and Physics. John Wiley & Sons, New Jersey. Table 9.5, p. 422] whereas the diffusion term is in the order of $(2D_0 \Delta t)^{1/2} \approx 10$ µm for $D_0 = 10^{-6}$ m²/s and $\Delta t = 100$µs. Hence, since the mean free path is much smaller, we argue that the diffusion controlled approach is still valid.*

The method employed in this study is not novel. If it were applied properly, it should give results identical to the ones from the solution of Smoluchowski problem since no new physics is introduce into the model.

*With the corrected model we now observe clusters with sizes above 0.85 nm. Figure 1 of this reply shows the size distribution after 50 s for a temperature of 200 K. We now calculate the nucleation rate based on counting the actual particle number.*

I do not recommend the manuscript for publication in the journal Atmospheric Chemistry and Physics.

*Figure 1: The size distribution of all particles after 50 s for n=10$^7$ cm$^{-3}$ and T = 200 K.*

[Figure]

---

## Author Comment (AC4) · 22 Sep 2017

FMs. Ref. No.: ACP-2017-269

Title: A 3D particle Monte Carlo approach to studying nucleation

*We thank the referees for their careful reading and useful suggestions. Our answers are inserted into the reports in italics.*

*Furthermore, we have clarified a few issues in the paper, added additional material and improved language and structure of the paper.*

**Reviewer #4**:

This paper attempts a Monte-Carlo study of the vapor-phase evolution of sulfuric acid molecules under clustering and evaporation of monomer. These species are treated in the paper as spheres with initial monomer size of 0.15nm and nucleation rate derived from the formation of clusters at 0.85nm. It is not clear what the critical nucleation cluster size is, but it is presumably smaller than 0.85nm. Initial gas-phase concentrations are at 10^7 and 10^8 sulfuric acid molecules per cc in an ambient atmosphere of pressure 1 bar.

*With the corrected model we now observe clusters with sizes above 0.85 nm. Figure 1 of this reply shows the size distribution after 50 s for a temperature of 200 K. We now calculate the nucleation rate based on counting the actual particle number.*

Under the simulated conditions the mean-free path is about 70nm, which is much larger than even the 0.85nm size considered for nucleation to occur. Therefore clusters are in the free-molecular size regime (mean free path much larger than the particle size). The diffusion controlled regime occurs when the mean free path is much smaller than the particle size. So it is not clear why the authors use the diffusion-controlled model of Fransaer and Penner, cited on page 12 line 14, which applies to Brownian-like diffusive motion of individual molecules or ions in dilute aqueous electrolytes. Unfortunately, this is a problem with the manuscript that affects the growth law throughout and makes it unacceptable for publication.

*The mean free path of sulphuric acid clusters in air is approx. 10-60 nm [J.H. Seinfeld and S.N. Pandis, 2006. Atmospheric Chemistry and Physics. John Wiley & Sons, New Jersey. Table 9.5, p. 422] whereas the diffusion term is in the order of $(2D_0 \Delta t)^{1/2} \approx 10 \ \mu m$ for $D_0 = 10^{-6} \ m^2/s$ and $\Delta t = 100 \mu s$. Hence, since the mean free path is much smaller, we argue that the diffusion controlled approach is still valid.*

Additional comments: The authors appear to be dealing with a much simpler problem than they treat: The clusters are treated as spheres and there is no explicit structure or inter-molecular potential assigned to them. The authors might as well be treating clusters in the capillarity approximation of nucleation theory, which treats clusters as spheres with sharp interface and no inherent structure other than that of the bulk liquid (for density) and a surface tension for evaporation rate from the Kelvin relation that is much like the author's Eq. 7.

*The referee is right in the sense that we do not include the structure or the intra-molecular potential of clusters. However, we do take into account the admixture of water as well as the cluster evaporation which depends on the strucutre of the clusters. In that case, we do include the strucutre implicity which is sufficient to determine the position of particles in 3D space as well as the size distribution.*

Such simplicity suggests that collisions might be tracked through Fokker-Planck type diffusion-drift coagulation/evaporation equations without knowing their precise positions in 3D – as only a detailed dynamics would require. Then MC could be used for calculation of arrivals and departures based statistically on cluster number densities and evaporation rates. Presumably the authors would end up with a Boltzmann distribution of cluster size. The reviewer is not an expert in molecular simulations but it does seem that detailed knowledge of 3D molecule/cluster position is overkill for spherically symmetric molecule/cluster interaction. Moreover, such detailed knowledge might be washed out anyway given the assumption of randomness that is already part of the MC method.

*It is true that there are other ways to obtain the size distribution of the clusters. However, the aim of the present manuscript is to introduce a new model to model nucleation and the evolution of the size distribution.*

[Figure]

Figure 1: The size distribution of all particles after 50 s for $n=10^7$ $cm^{-3}$ and $T = 200$ K.

[Figure]